# Pathogenesis of Alkali Injury-Induced Limbal Stem Cell Deficiency: A Literature Survey of Animal Models

**DOI:** 10.3390/cells12091294

**Published:** 2023-05-01

**Authors:** Lina Sprogyte, Mijeong Park, Nick Di Girolamo

**Affiliations:** Mechanisms of Disease and Translational Research, School of Biomedical Sciences, Faculty of Medicine and Health, University of New South Wales, Sydney, NSW 2052, Australia

**Keywords:** corneal alkali burn, limbal stem cell deficiency, pathogenesis

## Abstract

Limbal stem cell deficiency (LSCD) is a debilitating ocular surface disease that eventuates from a depleted or dysfunctional limbal epithelial stem cell (LESC) pool, resulting in corneal epithelial failure and blindness. The leading cause of LSCD is a chemical burn, with alkali substances being the most common inciting agents. Characteristic features of alkali-induced LSCD include corneal conjunctivalization, inflammation, neovascularization and fibrosis. Over the past decades, animal models of corneal alkali burn and alkali-induced LSCD have been instrumental in improving our understanding of the pathophysiological mechanisms responsible for disease development. Through these paradigms, important insights have been gained with regards to signaling pathways that drive inflammation, neovascularization and fibrosis, including NF-κB, ERK, p38 MAPK, JNK, STAT3, PI3K/AKT, mTOR and WNT/β-catenin cascades. Nonetheless, the molecular and cellular events that underpin re-epithelialization and those that govern long-term epithelial behavior are poorly understood. This review provides an overview of the current mechanistic insights into the pathophysiology of alkali-induced LSCD. Moreover, we highlight limitations regarding existing animal models and knowledge gaps which, if addressed, would facilitate development of more efficacious therapeutic strategies for patients with alkali-induced LSCD.

## 1. Introduction

The ocular surface is covered by two distinct epithelia of corneal and conjunctival identity which are separated by the limbus, an anatomical region that circumscribes the peripheral cornea (Figure 1a–d). Limbal epithelial stem cells (LESCs), which reside within the basal layer of the limbal epithelium and are supported by a highly specialized niche microenvironment, continuously replenish the aging or damaged corneal epithelia as regenerating cells migrate centripetally from the limbus towards the cornea’s apex [1].

Depletion or dysfunction of LESCs gives rise to a condition called limbal stem cell deficiency (LSCD), characterized by conjunctivalization of the cornea as the barrier function of the limbus is lost [2] (Figure 1e–m). Depending on the extent of limbal involvement, LSCD is categorized as partial with incomplete conjunctivalization, or total whereby the cornea is completely enshrouded with pathological conjunctival tissue [2]. 

The leading etiology of LSCD is chemical burns to the ocular surface [2]. In contrast to acids that bind to proteins on the outer corneal surface which form a protective barrier preventing further acid penetration, the hydroxyl ions present in alkaline substances induce saponification of fatty acids in cellular membranes, facilitating its rapid penetration deep into the cornea and dissolution of stromal collagen [3]. Therefore, extensive alkali burns that involve the limbal region not only deplete the LESC pool but can also destroy the underlying extracellular matrix (ECM) including limbal niche cells (LNCs). The consequential corneal conjunctivalization is accompanied by corneal neovascularization (CNV), chronic inflammation and stromal fibrosis [4]. 

Alkali injury-induced LSCD is the major indication for surgical LESC transplantation, which remains its definitive management strategy since no medical treatment currently exists that facilitates recovery of the depleted or absent LESC pool [5,6]. However, the success of a surgical stem cell intervention relies crucially on the level of co-existing inflammation and CNV. The same principal also applies to keratoplasty, which is a secondary intervention in patients with LSCD when residual stromal scarring significantly impinges on vision [6].

Better insight into the pathophysiological events that are initiated upon corneal alkali burn is necessary for the development of novel adjuvant medical therapies to improve surgical outcomes, or potentially to eliminate the need for a surgical intervention altogether. Corneal wound healing is a highly complex and dynamic process that involves reciprocal cellular interactions between epithelial, stromal, neuronal and immune cells. Activation of intracellular signaling cascades is largely mediated by a host of cytokines and growth factors (GFs) which regulate wound healing by initiating apoptosis, proliferation, migration, differentiation and ECM remodeling [7]. This review will discuss the current knowledge of the pathogenesis of alkali-induced LSCD in animal models, highlighting outstanding questions which, if addressed, would deepen our understanding of the disease process and open the door to new efficacious treatments for patients with unmet medical needs.

## 2. Models of Corneal Alkali Injury and LSCD

Animal models are pivotal to gaining valuable insights into the underlying pathogenesis of human disease. Numerous studies have explored pathophysiological mechanisms initiated in response to limbal-sparing central corneal alkali burns but fail to determine the effect of this type of injury on LESCs or the phenotype of re-epithelialized cells, given the primary objective is to study an animal model of inflammatory neovascularization and/or fibrosis, rather than LSCD. Nevertheless, several reports indicate that alkali burns of the central cornea heal without conjunctivalization [8,9].

The involvement of numerous signaling pathways has been elucidated in the pathogenesis of central corneal alkali burns [10,11,12,13,14,15,16,17,18,19,20,21,22]. Although these models provide important clues regarding the molecular mechanisms of alkali-induced LSCD, it is plausible that alkali injuries directly involving the limbus elicit a different or an additional set of pathophysiological responses, as this region is extensively vascularized and harbors LESCs together with various specialized LNCs, namely, limbal mesenchymal stem cells (MSCs) [23], immune cells [24], and melanocytes [25]. Notably, exogenous administration of LNCs in alkali-induced LSCD models [26] result in attenuated inflammation, opacification and CNV, highlighting the active participation of these cells and their role in modulating corneal healing. In cases of central corneal wounding, this is likely mediated by endogenous LNCs. Interestingly, limbal MSCs express markers of LESCs and have the capacity to transdifferentiate into corneal epithelia under culture conditions that simulate a native microenvironment [23], although in vivo evidence of such an identity change during physiological wound healing is lacking. The significance of epithelial–stromal interactions in the cornea and their role in pathological states has been emphasized [7], but it is unclear to what extent such interactions are dependent on the phenotype of the regenerated epithelia that cover the stroma following alkali burns (i.e., corneal versus conjunctival). Therefore, care should be taken when interpreting and evaluating cellular interactions and molecular signaling cascades in limbal-sparing versus limbal-encompassing alkali injuries.

For this review, animal models of alkali burn to the central cornea, where a direct limbal injury was not specified and corneal conjunctivalization was not confirmed will be referred to as a central alkali burn (CAB) model, as opposed to a limbal alkali burn (LAB) designed to induce LSCD and/or development of corneal conjunctivalization. Notably, LAB models demonstrate a huge variability in methodology for disease induction, although sodium hydroxide is by far the most used inflicting agent. However, its concentration, volume, mode of delivery, time of exposure and clearance method differ somewhat in each LAB study. Nonetheless, LAB models discussed in this review all achieved their goal of inducing LSCD, be it partial or total, highlighting some key pathological mechanisms that are shared across studies. Importantly, the clinical translational relevance of the LAB model depends less on the method of disease induction, but rather on the level of methodology detail stated and confirmation of resultant LSCD by well-established phenotypic markers [27].

## 3. Natural History and Pathophysiology of Alkali-Induced LSCD

Chronologically, the disease process begins the moment the alkali agent contacts the ocular surface (Figure 1 and Figure 2). Based on McCulley’s classification [28], it can be divided into four distinct stages: (i) immediate phase (0 to 24 h) during which pathological responses are initiated as a direct consequence of the chemical insult; (ii) acute stage (1 to 7 days) characterized by amplified inflammation, neovascularization and re-epithelialization; (iii) early reparative period (8 to 21 days) during which inflammation transitions from acute to chronic, and is associated with stromal fibrosis; and (iv) late reparative phase (>21 days) with multiple attempts by the cornea to remodel and heal.

### 3.1. Immediate Phase and Induction of Inflammation

The immediate phase of the disease process (0 to 24 h post-injury) begins when the alkali agent contacts the ocular surface and saponification of cell membranes promotes cellular damage and tissue necrosis (Figure 1e–g and Figure 2b). Histopathological evolution in CAB models involves activation of resident macrophages as early as 2 h after injury [29], followed by neutrophil infiltration from the limbal margin at 4–6 h post-injury [9,29,30], whilst influx of macrophages is minimal at this stage [29]. In a mouse LAB model, infiltration of polymorphonuclear leukocytes into the anterior stroma was confirmed 24 h post-injury [31].

Biochemically, CAB and LAB cause immediate release of reactive oxygen species (ROS), danger-associated molecular patterns (DAMPs) and cytokines from necrotic and damaged tissue (Figure 2b). In LAB, immediate increase in oxidative stress facilitates activation of the nuclear factor-kappa B (NF-κB) pathway [32], since NF-κB is a redox-sensitive transcription factor [33] which subsequently induces expression of numerous genes encoding pro-inflammatory cytokines (e.g., interleukin (IL)-1, IL-6) and chemokines [e.g., monocyte chemoattractant protein (MCP)-1, C-X-C motif ligand 1 (CXCL1)]. The molecular mechanisms regulating its transcriptional activation have been reviewed elsewhere [34,35]. In CAB, ROS levels are elevated at 2–6 h post-wounding [36] and coincide with increased transcription of leukocyte recruiting chemokines such as MCP-1 [9,29], macrophage inflammatory protein (MIP)-1α [29], IL-8 [9] and CXCL1 [29]. Pro-inflammatory cytokines IL-1β and IL-6 [9,29,36,37] were also produced by infiltrating leukocytes within the limbus, which displayed nuclear NF-κB. Interestingly, tumor necrosis factor (TNF)-α mRNA remained at baseline levels during the first 24 h following CAB [9,29,37]. Congruent with these results, a mouse LAB model displayed increased IL-6 but normal TNF-α levels 6 h after injury [38].

IL-1β activation requires cleavage of pro-IL-1β into its mature species [34], mediated by the nucleotide-binding oligomerization domain, leucine-rich repeat and pyrin domain containing protein 3 (NLRP3) inflammasome [35]. Its critical role was demonstrated in CAB studies, since reduced IL-1β levels were detected after pharmacological inhibition of NLRP3 [39], as well as in *Nlrp3*^−/−^mice [40]. Notably, it is recognized that IL-1β activates the NF-κB pathway [34], thereby promoting its own transcription as well as that for *Nlrp3*. This generates an inflammatory auto-amplification cascade responsible for rapid leukocyte recruitment and cytokine production.

Overall, these studies suggest that activation of NF-κB signaling via ROS and DAMPs, and signal amplification by activated IL-1β are early events that initiate and sustain the innate immune response upon corneal alkali injury.

### 3.2. Acute and Early Reparative Phases

The acute (1 to 7 days) and early reparative (8 to 21 days) phase of corneal healing upon alkali injury is a critical period in disease evolution (Figure 2c), featuring dynamic changes in inflammation, neovascularization, re-epithelialization and fibrosis.

#### 3.2.1. Inflammatory Milieu

In rodent models of LAB and CAB, influx of inflammatory cells that encroach from the periphery eventually extend to the central cornea over the first 7 days [9,29,41] (Figure 1h,i, arrowheads). Neutrophils and macrophages are predominantly recruited into the anterior and mid stroma [29,42], where the former outnumbering the latter [41,43,44]. Neutrophil levels peak between days 1 and 4 post-exposure [29,43,45], then sharply decline by day 7 [9,29,45]. In contrast, stromal macrophages accumulate during the first and second week post-injury [29,32,46], after which they gradually disappear [29].

This directional migration of leukocytes is mediated by chemokines (Figure 3). Studies which utilized LAB and CAB models all confirmed augmented MCP-1 [10,15,29,32,44,47,48,49], MIP-1α [29,49], MIP-2 [15,44], CXCL1 [29,44] and IL-8 [10,44,47,48] levels, which peak in the acute phase of corneal healing, then decrease during ensuing weeks. Neutrophil migration is primarily mediated by MIP-2 [45], CXCL1 [45] and IL-8 [44], while MIP-1α [50] and MCP-1 [10] recruit macrophages. Moreover, leukocyte influx into the injured stroma after LAB was also promoted by heightened levels of integrin ligands such as intercellular adhesion molecule (ICAM)-1 and vascular cell adhesion molecule (VCAM)-1 [31,51], which tether inflammatory cells to the vascular endothelial cell wall facilitating trans-endothelial migration.

At this stage, ongoing ROS production by neutrophils and macrophages is sustained by NADPH oxidases (NOX). CAB studies revealed critical roles for NOX2 and NOX4 [51,52,53] in maintaining oxidative stress and the chemotactic response, since their pharmacological inhibition [53] or genetic knockout [51] result in attenuated leukocyte infiltration and cytokine production.

Cytokines that are released after alkali exposure have overlapping pro-inflammatory, pro-angiogenic and pro-fibrotic actions. Overall, in both LAB and CAB models, pro-inflammatory cytokines that are elevated during the acute stage of corneal healing include IL-1α [29,31,37], IL-1β [29,44,49,54], IL-6 [29,44,49,54], and vascular endothelial growth factor (VEGF) [29,36,55], whereas TNF-α [21,29] and pro-fibrotic transforming growth factor (TGF)-β [13,43] accumulate as acute inflammation subsides (Figure 3). Notably, macrophages are the main cell type responsible for VEGF and TNF-α production [29]. Other cytokines consistently upregulated following alkali burns include IL-17 [15,38], pro-angiogenic fibroblast growth factor (FGF)-2 [42,50] and platelet-derived growth factor (PDGF) [36], anti-angiogenic disintegrin and metalloproteinase with thrombospondin motifs (ADAMTS) [43,56] and thrombospondins (TSPs) [43,56], as well as matrix metalloproteinase (MMP)-2, -9, -13 and -14 [36,42,53] that facilitate tissue remodeling during repair. In regions of the cornea that lack epithelial coverage, neutrophil degranulation and MMP discharge can promote stromal ulceration and corneal perforation [29].

Several pathways actively partake in sustaining inflammation of the cornea in CAB models (Figure 2c). For example, alkali exposure induces p38 mitogen-activated protein kinase (MAPK) signaling which activates the downstream effector MAPK-activated protein kinase-2 (MK2) to selectively augment IL-6, IL-1β, MIP-1α, ICAM-1, and VCAM-1 production, thereby significantly increasing leukocyte accumulation [16,57]. In addition, signal transducer and activator of transcription 3 (STAT3) signaling in alkali-induced corneal inflammation was highlighted when its pharmacological inhibition reduced VEGF, MMP-2, MMP-9 and TGF-β1 expression, possibly by attenuating MCP-1 production to decrease macrophage infiltration [18]. Furthermore, the inflammatory response to CAB was also mediated by mammalian target of rapamycin (mTOR) activation, since rapamycin (a well-known mTOR inhibitor) suppressed leukocyte infiltration [20] and reduced pro-inflammatory cytokine expression [13]. CAB studies have revealed upstream signals that simultaneously integrate into the mTOR pathway, such as phosphoinositide 3-kinase (PI3K)/protein kinase B (AKT) and extracellular signal-regulated kinase (ERK) 1/2 axes [13,20], to promote mTOR-dependent protein synthesis. Correspondingly, the IL-1/IL-1 receptor I (IL-1RI)/ERK signaling cascade can mediate the alkali-induced inflammatory response, as its pharmacological inhibition decreases leukocyte recruitment and reduces pro-inflammatory cytokine and chemokine levels [15].

Intrinsic anti-inflammatory mechanisms are also initiated to keep acute inflammation in check and to counter its destructive effects (Figure 3). For example, raised anti-inflammatory IL-10 has been documented during the acute stage of healing in CAB [15,29]. Moreover, peroxisome proliferator-activated receptor (PPAR)-α, -β, and -γ isoforms, which are known to exhibit anti-inflammatory properties by suppressing NF-κB signaling [10,30,41], are increased after CAB and their expression is detected in infiltrating leukocytes and regenerating basal corneal epithelial cells [10].

Taken together, the inflammatory milieu in acute and early reparative phase is a balance between pro-inflammatory and anti-inflammatory signals generated by interdependent and overlapping intracellular and intercellular signaling cascades.

#### 3.2.2. Neovascularization

CNV is characterized by de novo formation of capillaries that extend from the limbus into an otherwise avascular cornea. This complex process is accentuated by the inflammatory milieu, thereby disrupting the pro-angiogenic/anti-angiogenic balance [56] (Figure 4).

Histologically, CNV in LAB models is visible on day 3 post-injury [58], and during the ensuing weeks, it develops into a prominent vascular reaction that is predominantly confined to the anterior and middle layers of the stroma [59], where inflammation prevails (Figure 2c). This activity proceeds with vascular endothelial cell proliferation and active sprouting at the apex of growing blood vessels [55,60], concomitant with leukocytes adhering to neovessel walls [55]. The neovascular response is also accompanied by lymphangiogenesis [61].

The pathophysiological mechanisms that govern alkali-induced CNV directly or indirectly rely on the molecular signaling cascades that are initiated as part of the inflammatory response. Consequently, the extent of CNV closely correlates with the level of inflammation. For example, disruption of the NF-κB signaling pathway to dampen inflammation reduces CNV in both LAB [32] and CAB [15,41,62,63,64] models.

A key mediator of alkali-induced CNV is VEGF [17,29,36] (Figure 4), a factor primarily produced by macrophages [29]. The VEGF-leukocyte nexus further results in VEGF-mediated macrophage recruitment, generating a positive feedback loop known as the immune amplification cascade [19]. VEGF promotes both hemangiogenesis and lymphangiogenesis via regulating endothelial cell proliferation, differentiation, migration and blood vessel lumen formation [17]. Several VEGF species have been documented in CAB models [15,17,65]. Notably, VEGF-A and VEGF-B bind to VEGF receptors (VEGFR)-1 and VEGFR-2 expressed on vascular endothelial cells to drive hemangiogenesis, while VEGF-C and VEGF-D bind to VEGFR-2 and VEGFR-3 expressed on the lymphatic endothelial cells to facilitate lymphangiogenesis. In CAB models, VEGF and VEGFR expression are further promoted by leucine-rich α-2-glycoprotein-1 (LRG1) [65], PDGF [36,66] and stromal derived factor-1α (SDF-1α) [67]. Pro-angiogenic FGF-2 [42] and angiopoietin-1 and -2 [41] are also increased during the initial stages of a corneal alkali burn and their interplay with VEGF further entices endothelial cell proliferation.

Several studies that utilize CAB models have investigated the molecular pathways related to VEGF signaling and proposed that CNV develops via activation of the VEGFR-2-mediated STAT3/PI3K/AKT circuitry [17] (Figure 4). Similarly, pharmacological inhibition of VEGFR-2 blocks ERK1/2, c-Jun N-terminal kinase (JNK) and p38 MAPK signaling and attenuates CNV [14]. The development of CNV in this model is further dependent on the IL-6/STAT3/VEGF-A signaling cascade [19]. CAB also stimulates the PI3K/AKT axis resulting in activation of mTOR signal, which in turn upregulates VEGF production and augments CNV [20]. Moreover, blockade of the WNT/β-catenin pathway downregulates VEGF and suppresses CNV [22].

In addition to vascular endothelial cell proliferation and migration, angiogenesis also requires a stromal space for neovessels to expand. This process is facilitated by several members of the MMP family of enzymes that degrade fibrillar collagens. Numerous MMPs are upregulated in both LAB and CAB models, namely, MMP-2, -8, -9, -13, and -14 [36,42,53]. MMP-2 and MMP-9 degrade the epithelial basement membrane (BM) facilitating the release of sequestered pro-angiogenic factors such as VEGF and FGF-2 into the stroma [68] (Figure 4). Interestingly, MMP-2 and MMP-9 levels correlate with pro-angiogenic PDGF upregulation [66], while MMP-13 expression by stromal keratocytes is induced by VEGF via VEGFR-3 [69,70].

While VEGF levels peak during the acute phase of corneal healing and subside in the weeks following, progressive CNV unrelated to VEGF expression implies the action of other pro-angiogenic factors in neovessel formation and maintenance [29]. Notably, anti-VEGF treatment in CAB models fails to completely abrogate CNV [71]. Interestingly, while anti-VEGF treatment was more effective at suppressing CNV 7 days post-CAB compared with tacrolimus (a potent anti-inflammatory drug), the latter resulted in lower CNV levels at the 1-month follow-up [49]. Indeed, there is evidence to suggest that CNV in CAB is mediated by VEGF-independent pathways, including Sonic hedgehog signaling [72] and MK2/p38 MAPK. The latter pathway promotes CNV by downregulating anti-angiogenic pigment epithelium-derived growth factor (PEDF) and upregulating pro-angiogenic cytokines, including IL-6, IL-1β, MIP-1α, MCP-1, ICAM-1, and VCAM-1, but not VEGF [16]. In addition to PEDF, anti-angiogenic factors Ly6/uPAR related protein-1 (SLURP1) [64] as well as netrin-1 [46] and -4 [63] are also suppressed after CAB, to further enhance this process.

In summary, the inflammatory process disrupts the pro-angiogenic/anti-angiogenic equilibrium and promotes CNV via VEGF acting as the key pro-angiogenic mediator.

#### 3.2.3. Re-Epithelialization

The foremost objective following an acute corneal alkali burn is re-establishing the epithelial barrier to minimize infection and prevent stromal melting and perforation. Epithelial regeneration is initiated during the acute phase of healing (Figure 1h–j and Figure 2c). In LAB models, in vivo fate mapping of regenerating epithelia confirmed corneal conjunctivalization [73,74]. Upon inflicting a 360-degree limbal burn that spares the inner cornea, lineage tracing demonstrated the emergence of epithelial clonal stripes from the conjunctiva and their migration towards the cornea’s apex during healing [74]. In another study where the alkali solution was only applied to the temporal limbus, epithelial regeneration likely occurred from the unharmed neighboring nasal limbus, as no corneal conjunctivalization was detected [75]. Alternatively, re-epithelialization of the cornea may have arisen via de-differentiation of corneal committed cells (previously demonstrated in mechanical, but not chemical limbal injury [74]), although no direct evidence to support this proposition currently exists [75].

In LAB models, phenotypic assessment of the re-epithelialized cornea confirms induction of LSCD. This is validated by the loss of a well-established corneal epithelial differentiation marker cytokeratin (K)12 [73,76,77,78], whilst biomarkers of conjunctival squamous epithelia (K13 and K19 [73,79]) as well as conjunctival MUC5AC^+^ goblet cells (GCs) are present (Figure 1h–j). GCs appear as early as 7 days post-injury [58] and persist throughout the early reparative phase [76,77,79]. Notably, the onset of CNV precedes the appearance of GCs, which are often associated with vascularized areas [58], although the significance of this correlation is not fully understood.

Interestingly, rare islands of the K3/12-positive epithelium have been detected in a rabbit LAB model 12 days after the injury, although these cells displayed increased apoptotic activity [78]. Other studies that employ LAB models also report increased apoptotic epithelia throughout the regenerated corneal surface 7 days post-exposure [54], as well as heightened epithelial proliferation in the conjunctivalized cornea, which peaked after 10 days [77], indicating increased cell turnover in the healing epithelium.

A cascade of molecular events orchestrates keratinocyte migration, proliferation, differentiation and connectivity to neighboring cells and the BM during the re-epithelialization process. Unfortunately, there are few studies that have investigated signaling pathways which govern epithelial behavior following alkali exposure. In a LAB model where the injury is confined to the temporal limbus, epithelial tissue regeneration is regulated via the yes-associated protein (YAP)-dependent mechanotransduction [75]. In this paradigm, the repopulating limbal epithelia over the alkali-stiffened ECM displayed nuclear translocation of YAP, which promotes differentiation of these cells and loss of LESC phenotype [75]. The intracellular PI3K/AKT and MAPK/ERK pathways, which are regarded chief molecular cascades that influence cell proliferation, are activated to regenerate the epithelium after CAB [80]. Notably, pharmacological inhibition of TNF-α/JNK [12] and mTOR [20] signals accelerates keratinocyte proliferation in CAB models, highlighting their role in re-epithelialization. Treatment with topical keratinocyte growth factor (KGF)-2 and FGF-2 also accelerates migration of corneal fibroblasts and fast-track re-epithelialization, likely by impinging on p38 MAPK and ERK1/2 pathways [81]. IL-6 can further promote corneal epithelial cell migration after CAB [37], possibly via the STAT3 pathway [82].

Overall, the precise re-epithelialization dynamics and molecular events responsible for corneal epithelial recovery after alkali burn remain an enigma, but one worthy of further exploration.

#### 3.2.4. Innervation

Corneal innervation is derived from the ophthalmic branch of the trigeminal nerve. Nerve trunks enter the cornea at the level of the stroma and pass horizontally. Occasionally, they deviate at right angles, pierce the BM and course beneath and between basal corneal epithelia as intraepithelial corneal basal nerves (ICBNs) [83] (Figure 5). In addition to their sensory function, corneal nerves are pivotal for maintaining ocular surface homeostasis, as well as modulating inflammation and wound healing [84]. The physical interaction between corneal nerves and epithelia is visualized by the near-perfect alignment of axonal extensions with epithelial migratory tracts, which together form an intricate whorl-like structure in the central cornea [85]. Functional interdependency between these two cell types is largely mediated by neurotransmitters and epitheliotropic and neurotropic factors (Figure 5a). For example, corneal nerves secrete epitheliotrophins such as substance P and calcitonin gene-related peptide (CGRP) which act to support renewal and repair of corneal epithelia [84]. Furthermore, corneal epithelia and keratocytes produce neurotrophins such as epidermal growth factor (EGF), nerve growth factor (NGF), glial cell line-derived neurotrophic factor (GDNF), brain-derived neurotrophic factor (BDNF) and neurotrophin-3 (NT3) and -4/5 (NT4) to facilitate nerve function and survival [84]. In addition, a mutually inhibitory interaction exists between corneal nerves and neovessels. For instance, selective trigeminal denervation results in rapid-onset CNV [86], whilst pharmacologically induced regional CNV promotes denervation [86].

Ocular surface injury with alkali agents directly destroys corneal nerves as they penetrate multiple layers of the cornea [87]. Such immediate and severe nerve injury triggers a complex process termed neurogenic inflammation, characterized by release of nerve-derived factors, stimulating leukocyte infiltration and activation [84]. This causes the release of pro-inflammatory mediators that further promote nerve damage, via another process called neuroinflammation [84]. This bidirectional interaction between neuropeptides and the pro-inflammatory milieu generates a positive feedback loop that perpetuates inflammation (Figure 5b).

The neuropeptide substance P is a key mediator of pathophysiological processes affecting corneal nerves. Its regulatory role in promoting cytokine and chemokine production, leukocyte recruitment, hemangiogenesis and corneal epithelial cell migration and adhesion during wound healing has been reviewed [84]. In a mouse CAB model, substance P production occurs early post-injury and primarily localizes to basal epithelia [88]. Genetically ablating [89] or pharmacologically antagonizing [61,88] this factor reduces leukocyte infiltration and attenuates hemangiogenesis and lymphangiogenesis. It is likely that substance P released from alkali-injured corneal nerves promotes hem- and lymphangiogenesis via the neurokinin-1 receptor (NK-1R) pathway [61,88].

In CAB models, corneal nerve regeneration is not fully achieved even 6 weeks after injury [87]. Nonetheless, the profound stromal nerve loss and diminished ICBN density gradually improves over time, although nerve architecture does not return to the pre-injury state, as demonstrated by increased axon beading and tortuosity [87]. Such morphological changes indicate that fibers attempt to regenerate and increase their metabolic activity [90,91]. Re-innervation is often associated with high incidence of ‘neuroma-like structures’ in the epithelial layer of the injured cornea [87]. However, these are likely active nerve fibers emanating from corneal stromal-epithelial nerve penetration sites [83] rather than pathological anomalies. Notably, in mechanically induced LSCD, nerve regeneration occurs but the pattern fails to return to its original spatial organization, instead, a chaotic arrangement forms [85].

It is well recognized that corneal nerve recovery positively correlates with accelerated epithelial wound healing in numerous animal models of disease [92,93]. However, the relationship between corneal re-innervation and epithelial wound healing in CAB or LAB models has not been characterized and therefore requires further elucidation.

#### 3.2.5. Fibrosis

Corneal fibrosis is a reparative response to stromal injury characterized by disruption of the regular collagen fibril arrangement and deposition of excessive and disorganized ECM by active repair cells known as myofibroblasts [94] (Figure 6). These cells are defined by their unique expression of α-smooth muscle actin (α-SMA) [95,96]. In response to alkali injury, any enduring resident keratocytes that escape apoptosis are activated to differentiate into corneal fibroblasts, which proliferate and migrate towards the wound site, while transforming into α-SMA^+^ myofibroblasts [9,11,29,95,96]. Importantly, the extent of myofibroblast generation and subsequent fibrosis is directly proportional to the severity of the alkali injury. This was recently demonstrated in rabbit CAB models which were generated using variable NaOH concentrations, whilst controlling its mode of application, duration of exposure and irrigation [97]. Myofibroblasts synthesize copious amounts of fibronectin [98] and collagen types I [29], III [9,41,99] and IV [99]. Notably, α-SMA and collagen type III are the principal biomarkers of stromal fibrosis [100].

Another established precursor to corneal myofibroblasts is bone marrow-derived fibrocytes, which migrate into the cornea from the limbal vasculature [101,102]. Notably, this source of myofibroblast generation has not been comprehensively investigated in CAB or LAB models, but has been characterized in other conditions that result in stromal scarring [101,102]. It is possible that LAB injuries which affect the whole corneal and limbal stroma lead profound keratocyte necrosis, resulting in bone marrow-derived fibrocytes as the main source for myofibroblast development. Given the phenotypic and functional differences between corneal fibroblast-derived and fibrocyte-derived myofibroblasts (e.g., the latter is more prone to excessive ECM deposition) [101], it would be interesting to determine any mechanistic differences in fibrotic response in CAB versus LAB burns.

Biochemically, corneal-fibroblast and bone marrow-derived fibrocyte transdifferentiation into myofibroblasts is largely mediated by the pro-fibrotic cytokine TGF-β [95,96] (Figure 6), which is consistently elevated after CAB and LAB [11,29,76,77,98]. Alkali injury to the epithelium and its BM allows TGF-β from tears and epithelial cells to penetrate into the stroma and drive myofibroblast development [101]. While TGF-β1 and TGF-β2 have well-established pro-fibrotic effects, an anti-fibrotic action has been ascribed to the TGF-β3 isoform [102,103]. This is intriguing given that all three isoforms can bind to the same TGF-β receptors and activate the same signal transduction pathways. Furthermore, TGF-β3 also upregulates the expression of many pro-fibrotic genes [102,103]. Perhaps, the specific effects of the TGF-β isoforms depends on the cytokine and GF milieu and other modulators that surround corneal stromal cells at a particular time during the corneal wound healing [101,102]. Notably, these have not been characterized in CAB or LAB models.

TGF-β is thought to act via both SMAD-dependent and SMAD-independent intracellular signaling pathways, which have been recently reviewed [94,96]. Notably, TGF-β/SMAD signal drives fibroblastic transformation of corneal endothelial cells, also known as endothelial-to-mesenchymal transition, which has been detected after CAB injury [104]. This fibrogenic reaction is responsible for the formation of a retrocorneal fibrous membrane [105]. The involvement of PI3K/AKT/mTOR cascade in corneal fibrogenesis was documented in a rabbit CAB model [21], and pharmacologically inhibiting mTOR signaling resulted in attenuated stromal fibrosis in a mouse CAB model [98]. While the PI3K/AKT/mTOR pathway can be activated by TGF-β, it acts independently from the SMAD network. Nonetheless, a crosstalk with the SMAD signaling axis may exist to further fine-tune pro-fibrotic circuits in CAB, as has been demonstrated in other fibrogenic disorders of the eye [94].

### 3.3. Late Reparative Phase

The late reparative phase (>21 days post-injury) involves continuous attempts by the cornea to remodel and self-resolve (Figure 2d). Despite the recognition that corneal wound healing upon alkali burn is a lengthy process, most animal studies which investigated the pathophysiology of such injury focused on events that take place during the acute and early reparative phases. There are scant details of what transpires beyond 1 month. Thus, comprehensive insights into the pathophysiological mechanisms that prevail long-term are highly desirable.

Examination of long-term histopathological features of LSCD corneas demonstrated significant increases in epithelial thickness mainly due to cellular hypertrophy, and to a lesser extent epithelial hyperplasia [106]. Intraepithelial edema with vesicular-like structures [106] likely occurs from a dysfunctional or damaged corneal endothelium, as these cells are replaced by a retrocorneal fibrous membrane, which forms adjacent to Descemet’s membrane and adheres to the iris forming peripheral anterior synechiae [107].

Ongoing corneal conjunctivalization during the late reparative phase has been evidenced by persistent K13^+^ and K19^+^ conjunctival epithelia and GCs, and lack of K3^+^/K12^+^ corneal epithelia [4,106] (Figure 1k–m). However, Kethiri and colleagues documented that despite inflicting a standardized LAB injury to induce total LSCD, a proportion of rabbits recovered some K3 expression at 5 months [106] and 9 months [4] post-injury. In most cases, K3^+^ epithelia were found in the central and mid-peripheral cornea [106]. Interestingly, some epithelial regions co-expressed K3, K19 and MUC5AC [4], indicating a mixed corneal and conjunctival epithelial identity. Although the cellular mechanisms leading to the restoration of corneal epithelial phenotype were not revealed, some clues were discovered. For example, LAB-injured corneas which recovered K3 expression also displayed a gradual decrease in opacity and CNV during the 9-month monitoring period. Histologically, this coincided with inflammatory cells and vascularization confined to the peripheral cornea [4].

It is possible that the recovery of K3^+^ epithelia in self-healing corneas originated from the limbal regions which retained LESCs that were not destroyed by LAB. Alternatively, the recovery of K3 expression in LAB models could be a result of transdifferentiation, whereby conjunctival epithelia switch to a corneal phenotype [108]. Interestingly, this phenomenon was recently reported in a mechanically induced LSCD mouse model [109]. Similarly, such epithelial transformation events could account for corneal keratinization (otherwise known as squamous metaplasia), with appearance of K10^+^ skin-like epithelia in LAB-injured rabbits [106].

Notably, the epithelial phenotype and transdifferentiation phenomenon (Figure 2d), if it actually occurs, is likely influenced by changes in ECM composition. At least, stromal biomechanics can determine the epithelial phenotype by supporting the LESC niche [75]. It is plausible that in LSCD, epithelial changes into cornea-like counterparts are enabled by concurrent stromal ECM remodeling, although long-term progressive improvement in collagen organization was only reported in a rabbit CAB model [100] and has not been investigated in LAB. The cellular interactions and molecular programs transpiring at the epithelial–stromal interface need further elucidation.

## 4. Conclusions and Future Directions

Alkali-induced LSCD is a complex disease manifesting as corneal conjunctivalization, chronic inflammation, CNV and fibrosis. While animal models of CAB assist in elucidating some pathophysiological mechanisms that drive inflammation, CNV and fibrosis, they fail to exhibit the cardinal pathological feature of LSCD, i.e., corneal conjunctivalization. Nonetheless, most CAB studies did not investigate the cellular phenotype of the ocular surface, and rarely was monitoring conducted beyond 1 month post-injury, leaving the possibility that LSCD could potentially develop in a severe CAB model, or at least some degree of LESC malfunction could co-exist, especially given the harsh microenvironment that ensues on the ocular surface following an exposure to alkali substance.

Although corneal alkali burn is the most common etiology of LSCD in humans [2], only a limited number of studies employed LAB models designed to recapitulate a full spectrum of pathological features characteristic of alkali-induced LSCD, including conjunctivalization. Of these, very few investigated disease progression with long-term monitoring or attempted to elucidate the signaling pathways accountable for pathogenesis, such as events transpiring at the epithelial–stromal or epithelial–neuronal interface. Today, cellular interactions and molecular signaling pathways which govern epithelial behavior following LAB remain a mystery. Further exploration of the relationship and pathophysiological programs initiated between the regenerating epithelium, corneal nerves and stromal components after LAB injury would enable the identification of new therapeutic targets. Should the notion regarding the cornea’s intrinsic capacity to elicit a self-healing program be confirmed in LAB models, delineating these signals and executing these programs would be instrumental for the development of new pharmacological, biological and/or cell-based therapeutics that could promote such a radical change. In the meantime, determining the most effective therapeutic strategies to halt progression of alkali-induced LSCD and improve the engraftment and regenerative potential of a surgical LESC transplant, which remains the gold standard curative treatment, is an important task for researchers to focus on.

## Figures and Tables

**Figure 1 cells-12-01294-f001:**
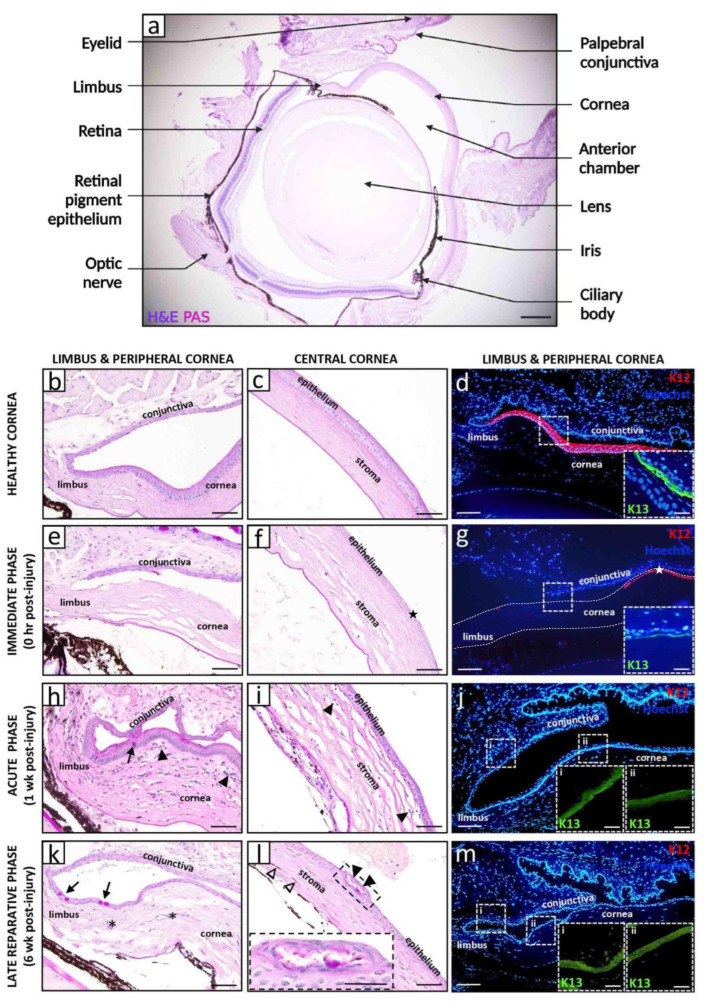
**Histopathological and immunophenotypic features of alkali-induced LSCD.** Representative images of a healthy (**a**–**d**) and alkali-induced LSCD (**e**–**m**) mouse cornea. LAB injuries were generated by applying a 2 μL drop of 0.25 M NaOH solution to the right cornea and limbus of male and female adult C57BL/6 mice (*n* = 12) for 30 s, followed by immediate irrigation with 30 mL 0.9% NaCl over 30 s. A cross-section of a healthy eye stained with hematoxylin and eosin (H&E) and periodic acid schiff (PAS) provides an overview of the tissue architecture (**a**). The limbal and central corneal regions are magnified in panels (**b**,**c**), respectively. The limbal region (**d**) is immunostained for corneal (rabbit anti-K12 [Abcam]; red, main panel) and conjunctival (goat anti-K13 [Santa Cruz Biotechnology]; green, insets) epithelial markers with Hoechst 33,342 (Life Technologies; blue) counterstain. Immediately after alkali burn (**e**–**g**), the limbal and corneal epithelia become necrotic and are shed ((**f**,**g**), stars), while the stroma appears less tightly compacted with loss of resident keratocytes. During the acute phase (**h**–**j**), K12^+^ corneal epithelia are replaced by K13^+^ conjunctival counterparts (**j**), concomitant with the appearance of GCs ((**h**), arrow). A heightened inflammatory response is evidenced by an influx of polymorphonuclear cells from the cornea’s periphery, venturing towards the center, predominantly within the anterior and middle stromal tiers ((**h**,**i**) arrowheads). The late reparative phase (**k**–**m**) is characterized by ongoing corneal conjunctivalization with persistence of GCs ((**k**), arrows) and associated CNV ((**k**), asterisks). The epithelium occasionally develops intraepithelial structures that contain PAS^+^ mucinous material ((**l**), arrowheads, magnified in inset). In some instances, LAB injury caused iridial adhesions to the posterior cornea and migration of iris-derived pigmented cells into the stroma ((**l**), hollow arrowheads). Scale bars represent 250 µm (**a**), 50 µm (**b**,**c**,**e**,**f**,**h**,**i**,**k**,**l**), 25 µm ((**l**) inset), 100 µm (**d**,**g**,**j**,**m**) and 25 µm ((**d**,**g**,**j**,**m**) insets).

**Figure 2 cells-12-01294-f002:**
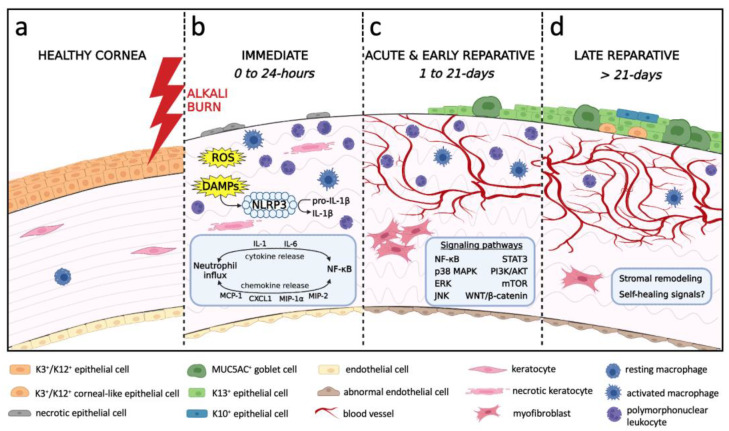
**Chronological schematic representation of the processes leading to the development of alkali-induced LSCD**. (**a**) Healthy cornea with intact corneal epithelium and endothelium, stromal keratocytes and resting macrophages. (**b**) Immediate phase (0 to 24 h) begins as the alkali agent penetrates through the cornea causing cellular damage, tissue necrosis, neutrophil infiltration and activation of resident macrophages. Release of ROS, DAMPs, cytokines and chemokines mediates neutrophil influx and NF-κB signaling. DAMPs trigger the assembly of NLRP3 inflammasome to cleave pro-IL-1β into IL-1β. Corneal edema results from hydrolysis of glycosaminoglycans and endothelial damage. (**c**) Acute and early reparative phases (1 to 21 days) are characterized by re-epithelialization with conjunctival epithelia, inflammation, angiogenesis and myofibroblast-mediated fibrosis. Activated signaling pathways include NF-κB, p38 MAPK, ERK, JNK, STAT3, PI3K/AKT, mTOR, WNT/β-catenin. The endothelium is replaced by a retrocorneal fibrous membrane. (**d**) Late reparative phase (>21 days) denotes an unsuccessful attempt by the cornea to remodel and self-heal. The epithelium consists of a mixture of GCs, K13^+^ conjunctival epithelia, K3^+^/K12^+^ corneal-like epithelia, and K10^+^ cutaneous-like squamous epithelia. Chronic inflammation and CNV persist. Stromal remodeling is evident by improved collagen organization and is associated with fewer myofibroblasts.

**Figure 3 cells-12-01294-f003:**
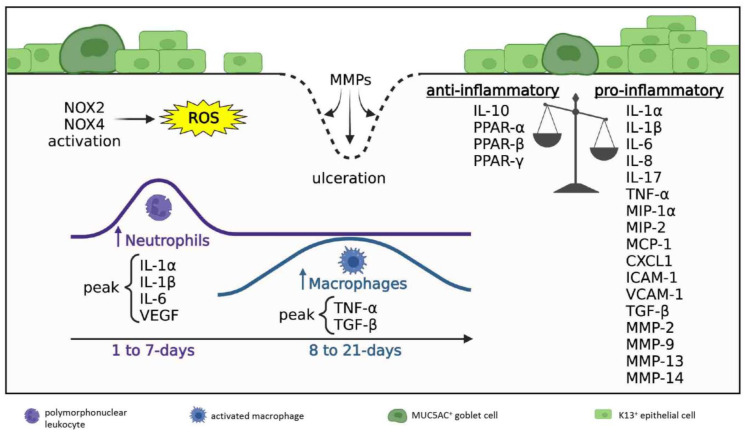
**Inflammatory milieu in acute and early reparative phases; 1 to 21 days post-injury.** Disrupted anti-inflammatory/pro-inflammatory balance where pro-inflammatory signals prevail. Ongoing ROS production is generated by NOX2 and NOX4 activity. Neutrophil density peaks during days 1–7 post-injury (acute phase), macrophage numbers peak during days 8–21 post-injury (early reparative phase). Neutrophils predominate and always exceed macrophages in number. The cornea is re-epithelialized by K13^+^ conjunctival epithelial cells and GCs. Regions lacking epithelial coverage are at risk of ulceration due to MMP-mediated collagen degradation.

**Figure 4 cells-12-01294-f004:**
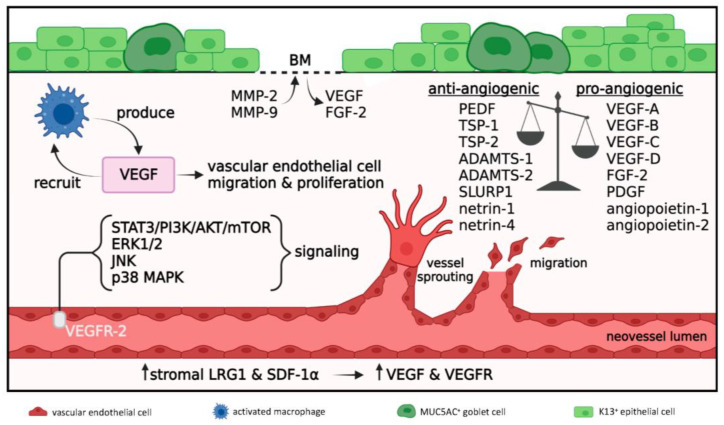
**Angiogenesis in acute and early reparative phases; 1 to 21 days post-injury.** Disrupted anti-angiogenic/pro-angiogenic balance where pro-angiogenic signals prevail. Macrophages produce VEGF which further promotes their recruitment and angiogenesis via vascular endothelial cell migration and proliferation. VEGF acts on VEGFR-2 expressed on vascular endothelial cells to activate STAT3/PI3K/AKT/mTOR, ERK1/2, JNK and p38 MAPK signaling pathways which promote angiogenesis. MMP-2 and MMP-9 degrade the epithelial BM resulting in release of sequestered VEGF and FGF-2. Increased levels of stromal LRG1 and SDF-1α further enhance VEGF and VEGFR expression to amplify angiogenesis.

**Figure 5 cells-12-01294-f005:**
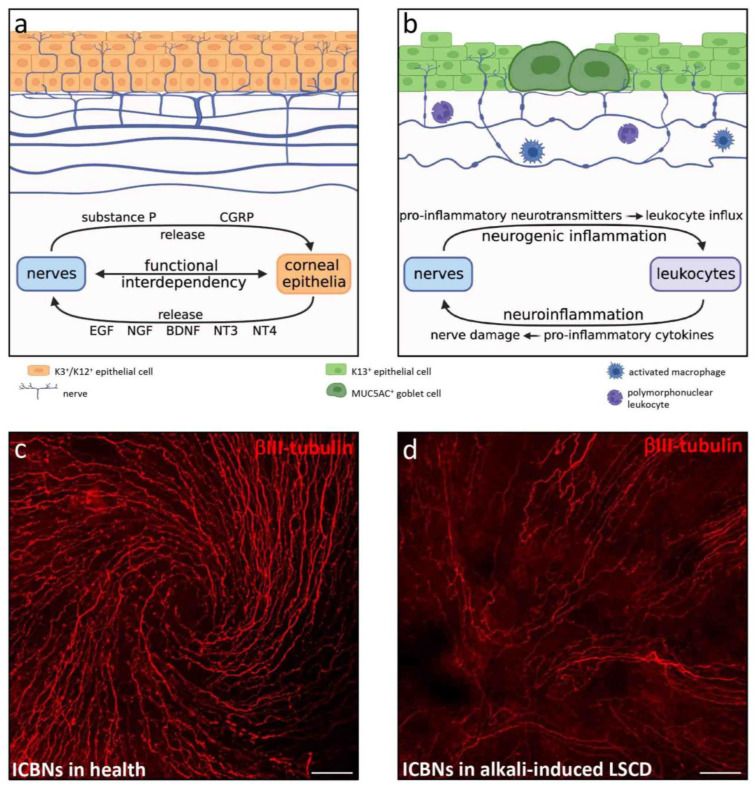
**Corneal nerves in health and in alkali-induced LSCD.** (**a**) Schematic depiction of a healthy cornea, where stromal nerve trunks pass horizontally and occasionally deviate at right angles to pierce the BM and become ICBNs that course beneath and between basal corneal epithelia. Nerve fibers in suprabasal epithelial tiers terminate as free nerve endings. Functional interdependency between corneal nerves and epithelia are mediated by epitheliotropic and neurotropic factors. (**b**) Schematic depiction of alkali-induced LSCD with stromal nerve loss, diminished ICBN density, increased axon beading and tortuosity. Alkali-injured nerves release pro-inflammatory factors to stimulate leukocyte recruitment (neurogenic inflammation), while recruited leukocytes secrete pro-inflammatory cytokines that cause further nerve damage (neuroinflammation). (**c**,**d**) Images of mouse central cornea immunostained with rabbit anti-βIII-tubulin (Sigma-Aldrich, Burlington, Massachusetts, USA), the pan-neuronal marker. (**c**) In the healthy cornea, ICBNs form a typical whorl-like pattern. (**d**) LAB injury (see legend to Figure 1) causes profound loss of ICBNs and a change in ICBN morphology concomitant with loss of the whorl. Scale bars (**c**,**d**) represent 50 µm.

**Figure 6 cells-12-01294-f006:**
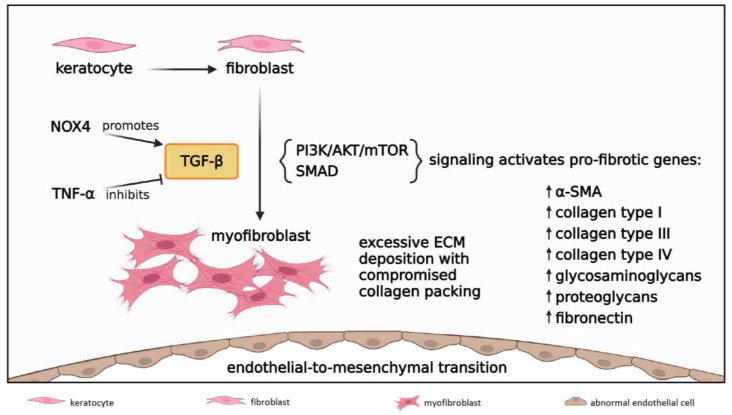
**Fibrosis in acute and early reparative phases; 1 to 21 days post-injury.** Keratocytes differentiate into fibroblasts which migrate to the wound site and further transform into myofibroblasts under the influence of TGF-β. NOX4 signal increases TGF-β levels while TNF-α inhibits TGF-β activity. TGF-β induces PI3K/AKT/mTOR and SMAD signaling cascades to initiate transcription of TGF-β-responsive pro-fibrotic genes, resulting in excessive ECM deposition with compromised collagen packing. TGF-β/SMAD signaling also drives endothelial-to-mesenchymal transition to form a retrocorneal fibrous membrane.

## Data Availability

The data presented in this study are available on request from the corresponding author. The data are not publicly available due to privacy.

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
