# Peer review of "Pathogenesis of Alkali Injury-Induced Limbal Stem Cell Deficiency: A Literature Survey of Animal Models"

_cells, 2023, doi:10.3390/cells12091294_

Round 1
Reviewer 1 Report
This is an excellent and detailed review on the effects and effectors behind corneal alkaline burns, with the emphasis on limbal stem cell deficiency that frequently occurs during these insults and exacerbates the condition. The pictures nicely illustrate complex events at the molecular and cellular level. Overall, this is a textbook review that should be useful for a very large audience of wound healing and ocular stem cell specialists.
This reviewer has only some minor concerns about the paper.
1. In Fig. 4, bFGF and FGF-2. are the same. Please keep FGF-2.
2. In Fig. 6, the authors might like to specify the profibrotic effects of TGF-beta 1 and 2 and anti-fibrotic effects of TGF-beta 3.
3. In terms of “keratocyte transdifferentiation into myofibroblasts”, there is still some debate about the origin of myofibroblasts. Are they the progeny of keratocytes or activated fibroblasts? It could be the latter, because epithelial removal usually causes massive keratocyte apoptosis. The authors might like to discuss this issue briefly.
4. The authors might like to expand the description of human corneal burns and therapeutic approaches in the Introduction. To this end, some citations may be needed, e.g., Saghizadeh M, Kramerov AA, Svendsen CN, Ljubimov AV. Concise review: stem cells for corneal wound healing. Stem Cells. 2017;35(10):2105-2114. PMID: 28748596;
5. Additionally, they might like to cite the following recent papers:
a. Villabona-Martinez V, Sampaio LP, Shiju TM, Wilson SE. Standardization of corneal alkali burn methodology in rabbits. Exp Eye Res. 2023 Mar 20:109443. PMID: 36948438.
b. Sun Z, Zhang M, Wei Y, Li M, Wu X, Xin M. A simple but novel glycymicelle ophthalmic solution based on two approved drugs empagliflozin and glycyrrhizin: in vitro/in vivo experimental evaluation for the treatment of corneal alkali burns. Biomater Sci. 2023 Feb 13. doi: 10.1039/d2bm01957d. PMID: 36779571.
c. Zhou J, Ding Y, Zhang Y, Zheng D, Yan L, Guo M, Mao Y, Yang L. Exosomes from bone marrow-derived mesenchymal stem cells facilitate corneal wound healing via regulating the p44/42 MAPK pathway. Graefes Arch Clin Exp Ophthalmol. 2023;261(3):723-734. doi: 10.1007/s00417-022-05956-4. PMID: 36576571.
6. Please briefly cover acid and thermal burns as well.
Reviewer 2 Report
1. Limbus circumscribes the peripheral cornea. It is better to show related data indicating how much percentage of corneal limbal injury can reach the threshold of LSCD, leading to CNV, re-epithelialization, fibrosis and even innervation, respectively.
2. All figures, including schematic representation and histopathology images, are good. I believe that these were all done by the authors, however, if any one of them was quoted from other reference sources, it should be marked.
